# The Impact of Nordic Walking on Bone Properties in Postmenopausal Women with Pre-Diabetes and Non-Alcohol Fatty Liver Disease

**DOI:** 10.3390/ijerph18147570

**Published:** 2021-07-16

**Authors:** Xiaming Du, Chao Zhang, Xiangqi Zhang, Zhen Qi, Sulin Cheng, Shenglong Le

**Affiliations:** 1Department of Orthopaedics, Shidong Hospital Affiliated to University of Shanghai for Science and Technology, Shanghai 200433, China; duxiaming@usst.edu.cn; 2Exercise Translational Medicine Center, Shanghai Jiao Tong University, Shanghai 200240, China; xiangqizhang@sjtu.edu.cn (X.Z.); qizhen10@sjtu.edu.cn (Z.Q.); shulin.cheng@jyu.fi (S.C.); 3Xidu Community Health Service Center of Fengxian District, Shanghai 201400, China; liwenying26@163.com; 4School of Life Sciences and Biotechnology, Shanghai Jiao Tong University, Shanghai 200240, China; 5School of Pharmacy, Shanghai Jiao Tong University, Shanghai 200240, China; 6Faculty of Sport and Health Sciences, University of Jyväskylä, 40014 Jyväskylä, Finland; 7The Key Laboratory of Systems Biomedicine, Ministry of Education, Shanghai Center for Systems Biomedicine, Shanghai Jiao Tong University, Shanghai 200240, China

**Keywords:** Nordic walking, postmenopausal women, fatty liver disease, bone markers, bone mineral density

## Abstract

This study investigated the impact of Nordic walking on bone properties in postmenopausal women with pre-diabetes and non-alcohol fatty liver disease (NAFLD). A total of 63 eligible women randomly participated in the Nordic walking training (AEx, *n* = 33), or maintained their daily lifestyle (Con, *n* = 30) during intervention. Bone mineral content (BMC) and density (BMD) of whole body (WB), total femur (TF), femoral neck (FN), and lumbar spine (L2-4) were assessed by dual-energy X-ray absorptiometry. Serum osteocalcin, pentosidine, receptor activator of nuclear factor kappa-B ligand (RANKL) levels were analyzed by ELISA assay. After an 8.6-month intervention, the AEx group maintained their BMC*_TF_*, BMD*_TF_*, BMC*_L2−4_*, and BMD*_L2−4_*, and increased their BMC*_FN_* (*p* = 0.016), while the Con group decreased their BMC*_TF_* (*p* = 0.008), BMD*_TF_* (*p* = 0.001), and BMD*_L2−4_* (*p* = 0.002). However, no significant group × time interaction was observed, except for BMD*_L2−4_* (*p* = 0.013). Decreased pentosidine was correlated with increased BMC*_WB_*(*r* = −0.352, *p* = 0.019). The intervention has no significant effect on osteocalcin and RANKL. Changing of bone mass was associated with changing of pentosidine, but not with osteocalcin and RANKL. Our results suggest that Nordic walking is effective in preventing bone loss among postmenopausal women with pre-diabetes and NAFLD.

## 1. Introduction

Type 2 diabetes (T2DM) is a widely prevalent chronic disease that affects bone health. Individuals with T2DM are known to have a higher risk for fractures with no change in or higher bone mineral density (BMD) than normal individuals [1]. In addition, studies have shown that non-alcoholic fatty liver disease (NAFLD) is related to decreased BMD in adults [2,3] and that NAFLD is negatively associated with BMD in postmenopausal women [4]. Diabetes, NAFLD, and osteoporosis metabolically alter the biology. Approximately one- to two-thirds of diabetic patients have NAFLD [5], and both diseases are known to be related to insulin resistance and bone metabolism. However, no studies have assessed bone properties in patients with comorbidities; hence, it is important to assess how bone properties would change after exercise intervention.

In humans, bone and glucose metabolism may share similar signaling pathways [6]. Early studies have shown that biomarkers such as osteocalcin (OC), glucose, and adipokines change with age but in a non-commensurate manner [7]. OC is a marker of bone formation and a bone matrix protein that is exclusively produced by osteoblasts and odontoblasts. OC plays a significant role as an endogenous insulin sensitizer [8]. The circulating levels of OC are known to increase with improved glycemic control in type 2 diabetes [8]. Receptor activator of nuclear factor kappa-B ligand (RANKL) is a marker of bone resorption and is accompanied by osteoprotegerin. Recent evidence has shown that RANKL is crucially implicated in the pathogenesis of T2DM [9]. In postmenopausal women with T2DM, the presence of NAFLD and clinically significant fibrosis was strongly associated with low RANKL levels [10]. In addition, pentosidine, a well-known advanced glycation end product (AGE), is an important surrogate marker for total AGE production. There is a negative correlation between serum pentosidine concentration and bone strength [11]. It has been reported that serum or urine levels of pentosidine positively correlate with fracture incidence and prevalence in T2DM [12]. Although the abovementioned biomarkers link both bone and glucose metabolism, the means by which those biomarkers are associated with bone properties in postmenopausal women with prediabetes and NAFLD are largely unknown.

Regular physical exercise has been recommended as an effective and safe non-pharmacological strategy to counter the aging-induced loss of BMD [13]. Nordic walking involves striding with the use of specially designed sticks and is a safe and relatively easy-to-learn form of fitness exercise. It is considered to be effective in patients with different chronic diseases [14], such as cardiovascular disease [15], aging [16], or women with breast cancer [17]. The movement sequences of Nordic walking make this physical activity suitable to support body posture and strengthen the muscles of the spine, shoulders, and hips [18]. Thus, we hypothesized that Nordic walking could be an ideal modality of exercise for postmenopausal women with pre-diabetes and NAFLD. It could help improve their BMD and bone turnover markers. To test our hypothesis, we evaluated the effects of an 8.6-month Nordic walking program on BMD by using dual-energy X-ray densitometry and biomarkers. This was considered to be associated with both bone and glucose metabolism in postmenopausal women with pre-diabetes and NAFLD.

## 2. Materials and Methods

### 2.1. Study Design and Participants

The present study is a part of a large interventional study that was published earlier [19,20]. In brief, the participants aged 50–65 years with an impaired fasting glucose level (IFG; between 5.6 and 6.9 mmol/L) or impaired glucose tolerance (IGT; between 7.8 to 11.0 mmol/L 2 h after the intake of 75 g glucose) and NAFLD (hepatic fat content > 5%) were considered eligible. In addition, women with serum follicle-stimulating hormone levels greater than 30 IU/L and last menstruation more than 6 months prior but within 10 years were included in this study. After eligibility was confirmed, the participants were randomly assigned (1:1:1:1) to 4 groups: aerobic exercise (AEx; *n* = 29), diet intervention (Diet; *n* = 28), aerobic exercise plus diet intervention (AED; *n* = 29), or no intervention (NI; *n* = 29). An intervention was carried out for an on average of 8.6 months (from 6 months to 11 months). For the purpose of this report, we focused on the effects of aerobic exercise on bone properties in postmenopausal women. Thus, we only included postmenopausal women and pooled the groups AEx and AED to form the AEx group (*n* = 33) and the groups Diet and NI to form the Con group (*n* = 30). Hence, we could increase the sample size, and the effects of the diet were comparable in both groups. 

The protocol of study was approved by the Ethics Committee of Shanghai Institute of Nutrition (No. 2013-003, 6 January 2013). This study conformed to the principles laid down in the world medical association Declaration of Helsinki for medical research involving human participants. Informed consent was obtained from all individual participants included in the study.

### 2.2. Exercise Intervention

During the 8.6-month intervention, all participants took part in supervised exercise sessions that were conducted 2–3 times per week in a community park that was close to their homes. Exercise sessions consisted of a 5 min warm-up and 5 min cool-down period (such as stretching and group exercises), and supervised progressive Nordic walking. The intensity and duration of exercises were increased from 60% to 75% of the maximum oxygen uptake (estimated from fitness tests) and from 30 to 60 min per session. The exercise intensity was monitored using a heart rate monitor (M5, Suunto, Vantaa, Finland).

### 2.3. Background Information

Background information regarding lifestyle as well as medical history was collected using questionnaires. Daily physical activity was recorded in an activity diary [19]. In addition, food records were collected to estimate the participant’s energy intake and intake of different nutrients during the study. The collected information was evaluated by exercise and nutrition experts.

### 2.4. Anthropometric and Bone Measurements

Height was determined using a wall-fixed measuring device, and body weight was determined using a calibrated scale, from which BMI was calculated. Dual-energy X-ray absorptiometry (DXA Prodigy, GE Lunar Corp., Madison, WI, USA) was used to assess whole-body lean mass and fat mass, as well as bone mineral content (BMC) and areal BMD of the whole body (WB), total femur (TF), femoral neck (FN), and lumbar spine (L2-4). The coefficient of variation for repeated measurements ranged from 0.9% to 1.3% for BMC and BMD. 

### 2.5. Clinical and Laboratory Measurements

Venous blood samples were taken in standardized fasting conditions between 7:00 a.m. and 8:00 a.m. Plasma samples were used to assess glucose, insulin, total cholesterol, high-density lipoprotein, low-density lipoprotein, triglycerides, and glycated hemoglobin A1c levels as explained previously [19]. The homeostasis model assessment of insulin resistance index was calculated using the formula: (fasting insulin concentration × fasting glucose concentration)/22.5 [21]. Glucose tolerance tests were performed after overnight fasting, and at 30 min, and 2 h after the intake of 75 g glucose for the assessment of serum insulin and glucose.

The serum concentration of OC as a bone formation marker was assessed by ELISA using Human Osteocalcin Quantikine ELISA kit (produced by R&D Systems, Minneapolis, MN, USA; assay sensitivity 0.898 ng/mL). RANKL was assessed using the Human TNFSF11 ELISA kit (Abcam, Cambridge, MA, USA, assay sensitivity 10 pg/mL). Pentosidine was assessed using the Human Pentosidine ELISA Kit (CUSABIO Technology LLC, Houston, TX, USA; assay sensitivity 7.81 pmol/mL).

### 2.6. Data Analysis

All analyses were performed using IBM SPSS statistics for Windows, version 25.0 (IBM Corp, Armonk, NY, USA). The Shapiro–Wilk test was used to check all data for normality. If data were not normally distributed, they were transformed by natural logarithm before further analysis. Descriptive statistics were used to present the data as means and 95% confidence intervals (95% CIs) unless otherwise stated. Student’s *t*-test was used to compare the differences at the baseline. Paired *t*-tests were used to analyze the changes with time in variables within groups.

Analysis of covariance for repeated measures (group × time) was used to assess the effects of the interventions by adjusting the corresponding baseline data, “years menopausal”, and intervention duration as the covariates. Measures of effect size in the analysis of covariance for repeated measures were shown in partial η^2^. 

Pearson correlation was used to assess the relationship between bone biomarkers and BMC/BMD at the baseline and post-intervention in the whole samples. Partial correlation coefficients, adjusted for “years menopausal” and intervention duration, were used to evaluate the changes of bone biomarkers with the changes of BMC/BMD from pre-to-post intervention in the whole samples. All tests were two-tailed, and a 5% probability level was considered significant.

## 3. Results

### 3.1. Baseline Participant Characteristics

The physical and clinical characteristics of the participants at baseline are summarized in Table 1. No significant difference was observed in any variable among the different groups.

### 3.2. Change in BMC and BMD after the Intervention

Comparisons of BMC and BMD at the different bone sites before and after intervention are shown in Table 2. After the 8.6-month intervention, the results showed a significant effect of group × time interaction (*p* = 0.013, partial η^2^ = 0.106) on the BMD*_L2-4_*. BMD*_L2-4_* levels were significantly lower in the CON group, while the AEx group preserved their earlier levels. Controlling for body weight and duration of the intervention, the results remained the same. No significant group × time interaction was observed in the other BMDs and BMCs. However, BMC*_FN_* was increased after intervention within the AEx group (*p* = 0.016, paired *t*-tests). In contrast, the CON group showed a decrease in their BMC*_TF_* (*p* = 0.008), BMD*_TF_* (*p* = 0.001) and BMD*_L2-4_* (*p* = 0.002) values after intervention, respectively (paired *t*-tests). 

### 3.3. Change in Bone Turnover Markers after the Intervention

Serum OC, RANKL, and pentosidine levels did not differ significantly between the AEx and CON groups both at the pre- and post-intervention (Figure 1). After the intervention, both the AEx and CON groups had decreased the levels of OC (*p* = 0.033 and 0.001, respectively, paired *t*-test) and pentosidine (*p* = 0.035 and 0.015, respectively, paired *t*-test), but no significant changes were observed in RANKL.

### 3.4. Associations between Biomarkers and BMC/BMD 

RANKL displayed a positive correlation with BMC*_WB_* (*r* = 0.398, *p* = 0.002), BMC*_TF_* (*r* = 0.396, *p* = 0.003), BMC*_FN_* (*r* = 0.320, *p* = 0.022), BMC*_L2-4_* (*r* = 0.322, *p* = 0.015), BMD*_WB_* (*r* = 0.506, *p* < 0.001), BMD*_TF_* (*r* = 0.454, *p* < 0.001), BMD*_FN_* (*r* = 0.421, *p* = 0.001), and BMD*_L2-4_* (*r* = 0.377, *p* = 0.004) values in the whole samples at the baseline, while these correlations disappeared after intervention (all *p* > 0.05, data are not shown). No significant associations were found between OC, pentosidine, and BMC/BMD at both pre- and post-intervention (all *p* > 0.05). Decreased pentosidine levels were associated with an increase in BMC*_WB_* (*r* = −0.352, *p* = 0.019, Figure 2) values. Even after adjusting for “years menopausal” and intervention duration, the significance remained (*r* = −0.369, *p* = 0.032). No correlations were observed between changes in OC and RANKL and variations in BMC/BMD at the different bone sites.

## 4. Discussion

In this study, we found that an 8.6-month aerobic exercise program that comprised Nordic walking chiefly, was able to maintain bone mass and density in patients with prediabetes and NAFLD. Previous studies have shown that the bone mass of the femur and lumbar spine decreases by approximately 1% annually at midlife and at an accelerated rate of 2% annually during the first few years after menopause in women [22,23]. In the present study, there was a ~0.4% increase in BMD*_L2-4_*, and BMD*_TF_* was conserved after 8.6 months of the exercise intervention when compared with a 2.6% decrease in the control group, indicating that the osteogenic effects were significant. This positive adaptation occurred in regions where a large portion of the postmenopausal women have osteoporosis or osteopenia.

Previous studies have shown that periodic exercise training in 1-year blocks (4–6-week blocks of high-intensity bone-specific exercise with intermittent moderate-intensity metabolism-specific exercise for 10–12 weeks) positively affected BMD at the lumbar spine that was assessed by peripheral quantitative computed tomography in early post-menopausal women with metabolic syndrome [24]. Skoradal et al. suggested that 30–60 min of soccer training twice a week for 16 weeks could effectively increase BMD (3.9%) in the lumbar spine in individuals 55–70 years of age with pre-diabetes [25]. Chien et al. reported that a 6-month graded treadmill walking program combined with stepping exercises using a 20 cm-high bench attenuated lumbar spine BMD loss in osteopenic postmenopausal women [26]. A 12-month walking program in early (≤6 years) postmenopausal women demonstrated a significant increase in BMD at the lumbar spine [27]. However, some studies also reported that walking did not increase BMD at the lumbar spine [28,29].

Nordic walking has characteristic diagonal movements with contralateral hand-foot coordination such that the swing phase is double (one leg and the pole in the opposite hand). Compared with walking without poles, it has different kinetic variables and involves stronger upper body movements [30]. Studies have shown that Nordic walking enhances muscular strength in healthy participants and in the elderly [14]. It is possible that muscle tension produces strains in the skeleton, which could induce bone formation [31]. We did not find significant increases in BMD of the lumbar spine and femur in the exercise group; however, there was a decrease in BMD in the control group (meaning amount of bone mass at the measured area of bone sites have changed, with bone becoming less dense). This indicates that our exercise intervention program could help maintain BMD.

OC is one of the bone turnover markers released during bone remodeling by osteoblasts or odontoblasts, which is believed to be associated with an increase in BMD. Studies have reported that exercise programs comprising football training, 40 min of jogging, and 20 min of gymnastics with wrist weights (0.8 kg on each arm) and strength training increased serum OC levels compared to controls [25,32,33]. On the contrary, we found no significant effects of Nordic walking on OC. Shibata et al. and Wochna et al. also demonstrated that exercise training did not increase OC levels while causing favorable changes in bone health [34,35].

RANKL was recently identified as an important cytokine that sustains osteoclast formation and survival [36]. We found that baseline BMC and BMD were associated with RANKL. However, Nordic walking did not change the serum concentration of RANKL. This is supported by previous studies in which there was no significant change in the serum RANKL levels after 8 months of combined exercise intervention in elderly participants [37,38]. In contrast, in one study among middle-aged men, high intensity (70–75% maximal heart rate) walking exercise was instructed for 10 weeks, five times per week. This led to a decrease in the serum concentration of RANKL when compared with moderate-intensity (50–60% maximal heart rate) exercise [39]. This suggested that RANKL signaling factors could be dependent on exercise intensity. 

AGEs, especially pentosidine, are considered to affect bone metabolism and contribute to bone fragility in patients with T2DM [40]. AGEs significantly inhibit osteoblast proliferation, differentiation, and mineralization and induce osteoblast apoptosis [41,42]. The formation of bone nodules in human osteoblasts was impaired by pentosidine [43]. In contrast, AGEs led to a decrease in osteoclast-induced bone resorption [44]. In our study, decreases in pentosidine levels were associated with increases in values of BMC of the whole body. Although we do not know the causal relationship, this association indicated that the change of serum pentosidine was correlated with the change of amount of bone mass. This result agreed with previous studies which have shown that pentosidine negatively correlated with bone strength [11] and was positively associated with fracture incidence in T2DM [12]. 

Our study has some limitations. First, the study only measured BMC/BMD using DXA. Bone quality in individuals with glucose impairment and NAFLD could be more important than BMD [45]. Second, the follow-up period was 8.6 months, which may not be long enough to observe the effects of exercise on BMC/BMD or bone turnover markers due to large individual variations. Third, all participants were from the same community, and the behavior of participants in the no-intervention group could have been influenced by those in the intervention groups, even though we asked them to maintain their existing lifestyle during the intervention. This is reflected by the observation that the fitness level of those in the non-intervention group also increased [20]. 

## 5. Conclusions

In conclusion, these findings indicate that brisk Nordic walking for 8.6 months is effective in preventing bone loss in the lumbar spine and femur among postmenopausal women with pre-diabetes. It could help counteract the impairment of bone metabolism in patients with comorbidities of prediabetes, such as NAFLD.

## Figures and Tables

**Figure 1 ijerph-18-07570-f001:**
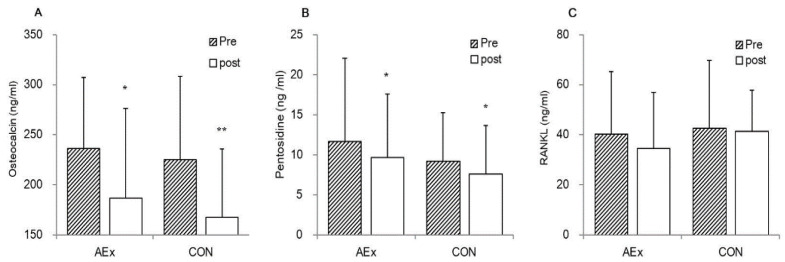
Bone turnover markers in AEx and CON pre-and post-intervention. Note: * *p* < 0.05, ** *p* < 0.01, paired *t*-tests were used to analyze the changes in variables within groups. Data are means ± SD (standard deviation).

**Figure 2 ijerph-18-07570-f002:**
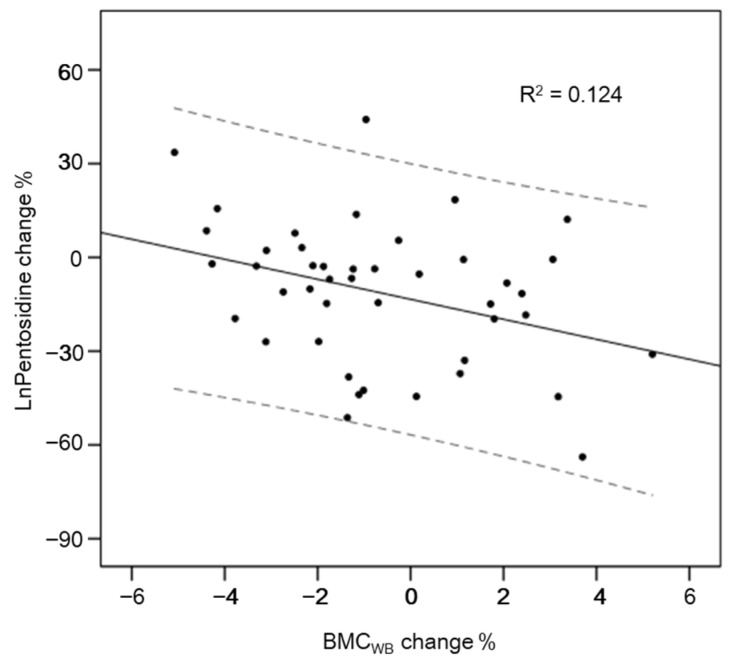
Correlation between pentosidine change and BMC*_WB_* change of pre-and post-intervention. Note: BMC*_WB_*, bone mineral content of the whole body.

**Table 1 ijerph-18-07570-t001:** Baseline characteristics of the exercise and control group.

	Exercise Group (*n* = 33)	Control Group (*n* = 30)
Age (years)	59.8 (58.5, 61.1)	59.7 (58.2, 61.1)
Height (m)	159.6 (157.6, 161.7)	157 (154.5, 159.5)
Weight (kg)	67.1 (64, 70.2)	63.8 (59.5, 68.1)
BMI (kg/m^2^)	26.4 (25.2, 27.6)	25.8 (24.4, 27.2)
Age at menopause (years)	49.1 (47.2, 51.0)	51 (49.4, 52.6)
Years post-menopausal (years)	10.6 (8.4, 12.9)	8.7 (6.8, 10.5)
FPG (mmol/L)	5.61 (5.39, 5.83)	5.55 (5.3, 5.81)
2hPG (mmol/L)	8.08 (7.47, 8.7)	8.20 (7.68, 8.72)
HbA1c (%)	6.12 (6, 6.24)	6.21 (6.09, 6.33)
FSH (nmol/L)	50.9 (41.5, 60.4)	50.1 (42.6, 57.5)
Osteocalcin (ng/mL)	236.4 (208.3, 264.4)	225 (189.8, 260.2)
Pentosidine (ng/mL)	13.7 (7.9, 19.4)	11.5 (6.1, 17.0)
RANKL (pg/mL)	46.4 (30.9, 62.0)	42.7 (32.6, 52.8)
Calcium intake (mg)	580.7 (452.2, 709.1)	666.2 (560.5, 771.9)
Physical activity (h/week)	2.45 (1.8, 3.1)	2.45 (1.83, 3.07)
Whole body T-score	−0.13 (−0.47, 0.22)	−0.20 (−0.54, 0.14)
Total femur T-score	−0.07 (−0.45, 0.31)	−0.20 (−0.61, 0.20)
Femoral neck T-score	−0.62 (−0.97, −0.26)	−0.54 (−0.93, −0.15)
Lumbar spine T-score	−0.13 (−0.65, 0.38)	−0.15 (−0.72, 0.42)

Note 1: Data are expressed as mean (95% confidence interval). Note 2: BMI, body mass index; FPG, fasting plasma glucose; 2hPG, 2-h plasma glucose; FSH, follicle-stimulating hormone; RANKL, Receptor Activator of Nuclear Factor-κ B Ligand. Note 3: Student’s *t*-test was used to evaluate the differences between groups at the baseline.

**Table 2 ijerph-18-07570-t002:** Pre- and post-intervention values of bone properties for the exercise and control groups.

	Exercise Group			Control Group			Time by Group
	Pre	Post	*p*	Pre	Post	*p*	*p*
Whole-body							
BMC (kg)	2.07 (1.98, 2.16)	2.06 (1.96, 2.16)	0.195	2.00 (1.89, 2.11)	1.99 (1.87, 2.10)	0.589	0.216
BMD (g/cm^2^)	1.07 (1.03, 1.11)	1.07 (1.03, 1.12)	0.961	1.07 (1.03, 1.11)	1.06 (1.02, 1.11)	0.756	0.952
Total-femur							
BMC (g)	29.2 (27.5, 30.9)	29.3 (27.6, 31.1)	0.927	28.1 (26.3, 29.9)	27.6 (25.8, 29.3)	0.008	0.587
BMD (g/cm^2^)	0.96 (0.91, 1.01)	0.96 (0.91, 1.02)	0.074	0.95 (0.89, 1)	0.93 (0.88, 0.98)	0.001	0.183
Femoral neck							
BMC (g)	4.02 (3.79, 4.25)	4.06 (3.83, 4.29)	0.016	3.98 (3.7, 4.25)	4.27 (3.93, 4.61)	0.131	0.579
BMD (g/cm^2^)	0.86 (0.82, 0.91)	0.87 (0.82, 0.92)	0.156	0.87 (0.82, 0.91)	0.86 (0.82, 0.91)	0.606	0.297
Lumbar spine							
BMC (g)	46.0 (43.1, 48.8)	45.8 (42.7, 48.9)	0.821	43.8 (40.5, 47.1)	42.7 (39.3, 46.1)	0.141	0.323
BMD (g/cm^2^)	1.12 (1.06, 1.18)	1.12 (1.06, 1.18)	0.592	1.11 (1.05, 1.18)	1.09 (1.02, 1.15)	0.002	0.013

Note 1: Data are shown as mean (95% confidence interval). Note 2: Bone properties were assessed by dual-energy X-ray absorptiometry. Note 3: BMC, bone mineral content; BMD, bone mineral density. Note 4: Paired *t*-tests were used to analyse the changes in variables within groups. Analysis of covariance for repeated measures (group × time) was used to assess the effects of the interventions by adjusting the corresponding baseline data, “years menopausal”, and intervention duration as the covariates.

## Data Availability

The datasets used and/or analysed during the current study are available from the corresponding author on reasonable request.

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
