# Peer review of "The Impact of Nordic Walking on Bone Properties in Postmenopausal Women with Pre-Diabetes and Non-Alcohol Fatty Liver Disease"

_ijerph, 2021, doi:10.3390/ijerph18147570_

Round 1
Reviewer 1 Report
General comment
In this study, the effects of Nordic walking on bone composition were investigated in 63 postmenopausal women: 33 for Nordic walking group and remaining 30 women for the control group. Pre- and post-intervention comparisons showed changes in several bone indices, however an interaction between time (pre- post) and group (walking and control) was statistically significant in BMDL2-4 in the ANOVA. Furthermore, serum pentosidine was negatively correlated with BMCWB. From these results, the authors concluded that Nordic walking is effective in preventing bone loss among postmenopausal women with pre-diabetes.
The reviewer considered that this manuscript is well organized and provides significant findings.
Specific comment
1. Two significant results were demonstrated in this study. First, the interaction between time (pre-post) and group (walking-control) was significant for BMDL2-4 in the ANOVA. Second, the decrease in pentosidine was associated with the increase in BMCWB. The former was the result for BMD and the latter was the result for BMC. How should we interpret this discrepancy? Please add some explanation for this if possible.
2. Consider to present the effect size (eta2 or partial eta2) of ANOVA.
3. L30-31: I think past tense is better, and consider to move this sentence in front of "Our results suggest that ......".
Author Response
Manuscript ID: ijerph-1280771
The Impact of Nordic Walking on Bone Properties in Postmenopausal Women with Pre-diabetes and Non-Alcohol Fatty Liver Disease
Xiaming Du , Chao Zhang , Xiangqi Zhang , Zhen Qi , Sulin Cheng , Shenglong Le *
We would like to thank the reviewer for the thoughtful comments and efforts that helped to improve our manuscript. In the following response, we address all comments and the correction according to the Reviewer's comments. All of our revisions have been marked up in the initial submission by using the “track changes” function.
Point 1: Two significant results were demonstrated in this study. First, the interaction between time (pre-post) and group (walking-control) was significant for BMDL2-4 in the ANOVA. Second, the decrease in pentosidine was associated with the increase in BMCWB. The former was the result for BMD and the latter was the result for BMC. How should we interpret this discrepancy? Please add some explanation for this if possible.
Response 1: Thank you for your constructive comments. There was no discrepancy regarding these two significant results. We apologias that we did not make it clear on these aspects.
BMDL2-4 is a variable to represent a given area of bone mass which was assessed from specific bone site (two demission), commonly called “areal bone mineral density” which indicates how dense the bone is at that area. BMC (bone mineral content) reflects how much bone mass is at measured sites (one demission). By using ANOCVA, we compared the intervention effect of AEx on bone variables and found that by contract to the Con group, the AEx group did not lose their areal BMDL2-4 (meaning amount of bone mass at the measured area of bone sites have changed that bone became less dense).
Regarding your second point, pentosidine (we measured from serum) is a biomarker which affects bone metabolism and contribute to bone fragility in patients with T2DM [40]. We found that decreased serum pentosidine was associated with increased BMCWB. Although we do not know the causal relationship, this association indicated that the change of serum pentosidine was correlated with the change of amount of bone mass. This result agreed with previous study which have shown a negative correlation between serum pentosidine concentration and bone strength [11]. To make it clear, we have modified the discussion to address your concern as follows:
Line 243 – 247:
“We did not find significant increases in BMD of the lumbar spine and femur in the exercise group; however, there was a decrease in BMD in the control group (meaning amount of bone mass at the measured area of bone sites have changed that bone became less dense). This indicates that our exercise intervention program could help maintain BMD.”
Line 271 – 276:
“In our study, decreased serum pentosidine was associated with increased BMC of the whole body. Although we do not know the causal relationship, this association indicated that the change of serum pentosidine was correlated with the change of amount of bone mass. This result agreed with previous studies which have shown that pentosidine negatively correlated with bone strength [11] and was positively associated with fracture incidence in T2DM [12]. “
Point 2: Consider to present the effect size (eta2 or partial eta2) of ANOVA.
Response 2: Thank you for your suggestion. We have added the description of partial eta2 in 2.6 data analysis and the partial eta2 of ANCOVA was shown in the section 3.2.
In 2.6. Data analysis line 151 – 152:
“Measures of effect size in the analysis of covariance for repeated measures were shown in partial η2.”
In Results line 171 – 173:
“After the 8.6-month intervention, the results showed a significant effect of group × time interaction (p = 0.013, partial η2 = 0.106) on the BMDL2-4.”
Point 3: L30-31: I think past tense is better, and consider to move this sentence in front of "Our results suggest that ......".
Response 3: Thank you for your suggestion. We have modified this as follows (line 29 - 32) :
“Changing of bone mass was associated with changing of pentosidine, but not with osteocalcin and RANKL. Our results suggest that Nordic walking is effective in preventing bone loss among postmenopausal women with pre-diabetes and NAFLD.”
Reviewer 2 Report
2021 07 05 Reviewer Report
I commend the authors on the completion of this manuscript. Overall it is well written and on an important topic. I have a few concerns highlighted below.
Line 85. This sentence is incomplete. I think it must be: “last menstruation more than 6 months ago, but within 10 years were 85 also included.”
Line 118. I think there is error. It must be “…a real…”
Line 149. Is it possible to explain this sentence a little more? What are the variables of interest? Was it made by group? In the whole sample? : “The Partial correlation coefficients, adjusted for “years menopausal” and intervention duration, were assessed to evaluate the correlation between pre-to-post changes and the variables of interest.”
Line 168: These results come from what statistical analysis?: “However, the AEx group revealed no changes in BMCTF, BMDTF, BMCL2- 4, and BMDL2-4 values, except increases in BMCFN (p = 0.016) values. In contrast, the CON group showed a decrease in their BMCTF (p = 0.008), BMDTF (p = 0.001)) and BMDL2-4 (p = 170 0.002) values.” I suppose they come from the “Paired t-tests used to analyze the changes in variables within groups.”, but it is not clear.
Line 178: I think this paragraph must be corrected in relation to table and figure. And the statistical test used must be included (I suppose they come from the “Paired t-tests used to analyze the changes in variables within groups): “Baseline plasma OC, RANKL, and pentosidine levels did not differ between the AEx and CON groups (Table 1).After the intervention, both the AEx and CON groups had decreased OC (p = 0.033 and 0.001, respectively) and pentosidine (p = 0.035 and 0.015, 181 respectively) levels, but no significant changes were observed in RANKL (Figure 1).”
Line 186: 3.4. Associations between biomarkers and BMC/BMD. These associations were studied in the whole sample? If yes, at baseline, it is correct, but decreases must be studied classified by group. Or it must be specified in point 3.3, that there are no differences between groups post intervention in OC, RANKL, and pentosidine levels.
Line 203. “Previous studies have shown that the bone mass of the femur and lumbar spine decreases by approximately 1% at midlife and at an accelerated rate of 2 % during the first few years after menopause in women.” Are these rates referred to a year?
Line 260. “Decreased pentosidine levels were associated with increased BMC of the whole body.” I think this sentence is not correct, because there is not increased BMC of the whole body in the sample. It is better to say: “Decreases in pentosidine levels were associated with increases in values of BMC of the whole body.”
Line 262: I think this sentence it is not correct: “Our results did not show that these three biomarkers were linked with bone and glucose metabolism. However, the same cannot be ruled out due to the limitations of this study.”, because some associations have been described. I think it is better to say only: No associations were found between the other biomarkers and BMC/BMD, but these results may be related, at least in part, to the limitations of this study.”
Finally, review abstract according to these corrections and add the hole conclusion in the abstract.
Author Response
Manuscript ID: ijerph-1280771
The Impact of Nordic Walking on Bone Properties in Postmenopausal Women with Pre-diabetes and Non-Alcohol Fatty Liver Disease
Xiaming Du , Chao Zhang , Xiangqi Zhang , Zhen Qi , Sulin Cheng , Shenglong Le *
We would like to thank the reviewer for the thoughtful comments and efforts that helped to improve our manuscript. In the following response, we address all comments and the correction according to the Reviewer's comments. All of our revisions have been marked up in the initial submission by using the “track changes” function.
Point 1: Line 85. This sentence is incomplete. I think it must be: “last menstruation more than 6 months ago, but within 10 years were 85 also included.”
Response 1: Thank you very much for the notice, we have now modified the sentence to make it complete as follows (line 86 - 88):
“In addition, women with serum follicle-stimulating hormone levels greater than 30 IU/L and last menstruation more than 6 months but within 10 years were included in this study.”
Point 2: Line 118. I think there is error. It must be “…a real…”
Response 2: Areal BMD is a special term for assessment of the bone mineral density in two dimensions.
Point 3: Line 149. Is it possible to explain this sentence a little more? What are the variables of interest? Was it made by group? In the whole sample? : “The Partial correlation coefficients, adjusted for “years menopausal” and intervention duration, were assessed to evaluate the correlation between pre-to-post changes and the variables of interest.”
Response 3: Thank you very much for your constructive comments and suggestions. To make it clearer, we have modified the text as follows (line 145 - 158):
“Descriptive statistics were used to present the data as means and 95% confidence intervals (95%CIs) unless otherwise stated. Student’s t-test was used to compare the differences at the baseline. Paired t-tests were used to analyze the changes with time in variables within groups.
Analysis of covariance for repeated measures (group × time) was used to assess the effects of the interventions by adjusting the corresponding baseline data, “years menopausal”, and intervention duration as the covariates. Measures of effect size in the analysis of covariance for repeated measures were shown in partial η2.
Pearson correlation was used to assess the relationship between serum bone biomarkers and BMC/BMD at the baseline and post-intervention in the whole samples. Partial correlation coefficients, adjusted for “years menopausal” and intervention duration, were used to evaluate the changes of serum bone biomarkers with the changes of BMC/BMD from pre-to-post intervention in the whole samples. All tests were two-tailed, and a 5% probability level was considered significant.”
Point 4: Line 168: These results come from what statistical analysis?: “However, the AEx group revealed no changes in BMCTF, BMDTF, BMCL2- 4, and BMDL2-4 values, except increases in BMCFN (p = 0.016) values. In contrast, the CON group showed a decrease in their BMCTF (p = 0.008), BMDTF (p = 0.001)) and BMDL2-4 (p = 170 0.002) values.” I suppose they come from the “Paired t-tests used to analyze the changes in variables within groups.”, but it is not clear.
Response 4: Thank you for your notice. Yes, these results were from Paired t-tests within groups. We have modified the texts to address your concern (line 176 - 179).
“However, BMCFN was increased after intervention within the AEx group (p = 0.016, paired t-tests). In contrast, the CON group showed a decrease in their BMCTF (p = 0.008), BMDTF (p = 0.001) and BMDL2-4 (p = 0.002) values after intervention, respectively (paired t-tests).”
Point 5: Line 178: I think this paragraph must be corrected in relation to table and figure. And the statistical test used must be included (I suppose they come from the “Paired t-tests used to analyze the changes in variables within groups): “Baseline plasma OC, RANKL, and pentosidine levels did not differ between the AEx and CON groups (Table 1).After the intervention, both the AEx and CON groups had decreased OC (p = 0.033 and 0.001, respectively) and pentosidine (p = 0.035 and 0.015, 181 respectively) levels, but no significant changes were observed in RANKL (Figure 1).”
Response 5: Thank you for this important comment and suggestion. We have modified the statistical analysis methods section to make it clear. In addition, we also added notes into footnote of Table and Figure legend.
In 2.6. Data analysis line 145 – 158:
“Descriptive statistics were used to present the data as means and 95% confidence intervals (95%CIs) unless otherwise stated. Student’s t-test was used to compare the differences at the baseline. Paired t-tests were used to analyze the changes with time in variables within groups.
Analysis of covariance for repeated measures (group × time) was used to assess the effects of the interventions by adjusting the corresponding baseline data, “years menopausal”, and intervention duration as the covariates. Measures of effect size in the analysis of covariance for repeated measures were shown in partial η2.
Pearson correlation was used to assess the relationship between bone biomarkers and BMC/BMD at the baseline and post-intrvention in the whole samples. Partial correlation coefficients, adjusted for “years menopausal” and intervention duration, were used to evaluate the changes of bone biomarkers with the changes of BMC/BMD from pre-to-post intervention in the whole samples. All tests were two-tailed, and a 5% probability level was considered significant.”
In Results line 167 - 168:
“Note 3: Student’s t test was used to evaluate the differences between groups at the baseline.”
And line 183 – 186:
“Note 4: Paired t-tests were used to analyse the changes in variables within groups. Analysis of covariance for repeated measures (group × time) was used to assess the effects of the interventions by adjusting the corresponding baseline data, “years menopausal”, and intervention duration as the covariates.”
And line 195:
“paired t-tests were used to analyse the changes in variables within groups.”
Point 6: Line 186: 3.4. Associations between biomarkers and BMC/BMD. These associations were studied in the whole sample? If yes, at baseline, it is correct, but decreases must be studied classified by group. Or it must be specified in point 3.3, that there are no differences between groups post intervention in OC, RANKL, and pentosidine levels.
Response 6: Thank you very much for your comments. Yes, it is whole sample. We have now followed your suggestion and modified the texts to make it clear:
In 3.3 line 188 - 192:
“Serum OC, RANKL, and pentosidine levels did not differ significantly between the AEx and CON groups both at the pre- and post-intervention (Figure 1). After the intervention, both the AEx and CON groups had decreased the levels of OC (p = 0.033 and 0.001, respectively, paired t-test) and pentosidine (p = 0.035 and 0.015, respectively, paired t-test), but no significant changes were observed in RANKL.“
In 3.4 line 198 - 204:
“RANKL displayed a positive correlation with BMCWB (r = 0.398, p = 0.002), BMCTF (r = 0.396, p = 0.003), BMCFN (r = 0.320, p = 0.022), BMCL2-4 (r = 0.322, p = 0.015), BMDWB (r = 0.506, p < 0.001), BMDTF (r = 0.454, p < 0.001), BMDFN (r = 0.421, p = 0.001), and BMDL2-4 (r = 0.377, p = 0.004) values in the whole samples at the baseline, while these correlations were disappeared after intervention (all p > 0.05, data are not shown). No significant associations were found between OC, pentosidine, and BMC/BMD at both pre- and post-intervention (all p > 0.05).”
Point 7: Line 203. “Previous studies have shown that the bone mass of the femur and lumbar spine decreases by approximately 1% at midlife and at an accelerated rate of 2 % during the first few years after menopause in women.” Are these rates referred to a year?
Response 7: Thank you for your comment, Yes, you are right, these rates referred to a year. To make it clear, we added in a word “annually” after the percentage as below (line 216 - 219):
“Previous studies have shown that the bone mass of the femur and lumbar spine decreases by approximately 1% annually at midlife and at an accelerated rate of 2 % annually during the first few years after menopause in women [22,23].”
Point 8: Line 260. “Decreased pentosidine levels were associated with increased BMC of the whole body.” I think this sentence is not correct, because there is not increased BMC of the whole body in the sample. It is better to say: “Decreases in pentosidine levels were associated with increases in values of BMC of the whole body.”
Response 8: Response: Thank you very much for your suggestion. Done (line 271 - 272).
Point 9: Line 262: I think this sentence it is not correct: “Our results did not show that these three biomarkers were linked with bone and glucose metabolism. However, the same cannot be ruled out due to the limitations of this study.”, because some associations have been described. I think it is better to say only: No associations were found between the other biomarkers and BMC/BMD, but these results may be related, at least in part, to the limitations of this study.”
Response 9: Thank you very much for your suggestion. We have modified the texts as follows (line 277):
“Our study has some limitations.”
Point 10: Finally, review abstract according to these corrections and add the whole conclusion in the abstract.
Response 10: Thank you very much for your suggestion. Done (line 29 - 32).